# The perceived impact of multiple sclerosis and self-management: The mediating role of coping strategies

**Maciej Wilski**[1]*, **Waldemar Brola**[2], **Magdalena Łuniewska**[1], **Maciej Tomczak**[3]

**1** Department of Adapted Physical Activity, Poznań University of Physical Education, Poznan, Poland,
**2** Collegium Medicum, Jan Kochanowski University, Kielce, Poland, **3** Department of Psychology, Poznań University of Physical Education, Poznań, Poland

* mwilski@wp.pl

**Data Availability Statement:** All relevant data are within the paper and its Supporting Information files.

## Abstract

Low level of self-management in people with multiple sclerosis (MS) is considered to be a predominant factor that leads to poor rehabilitation efficacy. Studies focusing on the relationship between self-management and psychological variables that can be modified could contribute to expanding the knowledge needed to propose interventional programs aiming at patient activation. This study aimed to analyze whether coping strategies play a mediating role in the association between the perceived impact of MS and level of self-management in people with MS. The cross-sectional study included 382 people with MS. The participants completed the Multiple Sclerosis Self-Management Scale—Revised, Multiple Sclerosis Impact Scale-29, and Coping Inventory for Stressful Situations. The study hypothesis was evaluated using mediation analysis. The STROBE checklist specifically prepared for cross-sectional research was applied in this study for reporting. Results indicate that the emotion- and problem-focused strategies of coping can be treated as mediating the association between the MS impact and level of self-management in people with MS. A negative relationship was found between the perceived MS impact and problem-oriented coping, while a positive relationship was found between problem-oriented coping and self-management. Furthermore, a positive relationship was found between the MS impact and emotion-oriented coping, while a negative relationship was found between emotion-oriented coping and self-management. The indirect role of avoidance-oriented coping was not significant. Our study confirms the role played by coping strategies in individuals' self-management. In MS, self-management determined by perceived MS impact can be controlled by decreasing emotional-coping while increasing problem-coping strategies. Our study imparts new knowledge regarding the potential interventions for improving the level of self-management in people with MS. It indicates that recognition of individuals' illness perceptions as well as maladaptive coping strategies can help health professionals identify those who might be having lower level of self-management.

**Funding:** The author(s) received no specific funding for this work.

**Competing interests:** The authors have declared that no competing interests exist.

## Introduction

Multiple Sclerosis (MS) is an unpredictable, difficult-to-control disease with treatment options focusing mainly on symptom relief and delaying disease progression. People with MS may not realize the benefits of costly, inconvenient, and painful treatment procedures because these are intended to prevent relatively uncommon but disruptive events of relapse and progression of disability over the years [1]. For this reason, a significant number of patients do not participate in treatment programs for at least a certain period [2], which causes delays in the assessment of treatment effectiveness. Therefore, increasing patient participation in therapeutic programs is highly necessary to plan the future course of action. This can be achieved by encouraging individuals to ensure active self-management of their disease. To date, patient participation is one of the greatest priorities of research conducted on MS [3].

Self-management corresponds to successful management of a disabling condition. It requires special skills supporting treatment through medication adherence, gathering information about illness and novel treatment options, participation in therapeutic decisions, performing self-care, maintaining social relationships, and finding emotional equilibrium [4, 5]. Patient involvement in the form of high self-management skills is not sufficient for achieving a successful outcome, but it increases the chances of achieving a positive outcome. Recent studies provide significant evidence about the health advantages brought by self-management in people with MS [6, 7]. It is associated with a reduction of fatigue [8], improvement in health-related quality of life [9], adherence to medication [10], and increase in physical functioning [11, 12]. In the systematic review of Rae-Grant et al. [13], treatment outcomes in MS were shown to improve with the efficacy of self-management.

As self-management is the one of the most important elements contributing to improving the effectiveness of MS treatment, the search for factors supporting its growth is one of the major research directions. So far, studies focused on determining the correlates of self-management have revealed self-efficacy, realistic timeline, and treatment control perception as salient correlates of successful self-management in people with MS [14]. Other studies have indicated that people with MS, who have low support and lack socioeconomic resources, are likely to have poor self-management [15]. Interestingly, no studies have dealt with the simple relationship between the patient's perceived state of the disease and his/her level of self-management, which is the core of the Common-Sense Model of Self-Regulation [16] emphasizing the role of health threat perception in regulating health behaviors.

Most self-regulatory theories assume that behavior is a consequence of personal experience [17]. When affected by a disease, individuals develop certain perceptions about their health and attempt understanding and managing the problem [18]. Deteriorating health influences the patients to take intensified actions aimed at improving their health condition. In a situation when the disease does not lead to visibly negative effects, the individual does not need to act. This basic relationship is true in the case of predictable and controllable diseases. However, the scenario is different in some diseases like MS, in which the course of illness varies in time as well as between patients, is difficult to control, and the effects of self-management may not be evident for a long time. In such cases, the relationship between the perceived state of the disease and the actions taken to improve it may be conditioned by other factors. On the basis of the Common-Sense Model of Self-Regulation, an individual's perception of illness will be associated with self-management activities through coping strategies that serve as mediating variables.

Most models distinguish three primary dimensions of coping: problem-oriented coping, which involves a direct attempt for changing a stressful circumstance; emotion-oriented coping, which aims at lowering emotional distress; and avoidance-oriented coping, which includes

searching for other people (social diversion) or keeping occupied with another task (distraction) [19]. These categories are not mutually exclusive. Although evidence shows that coping strategies are stable [20], specific factors, such as the impact of a chronic illness, may change the preferred coping strategy, and consequently, the behavior. In the case of MS, coping has been shown to mediate the interaction between illness beliefs, mental and physical well-being [21], executive function and psychosocial adjustment [22], fatigue and the quality of life [23], disability and psychosocial loss [24], or perception of disease control and mental health [25]. However, the mediation role of coping in determining self-management behaviors has not yet been investigated.

Therefore, our study aimed to examine the relationship between the perceived impact of MS and level of self-management and to what extent the coping strategies act as mediators in this association. Since research regarding this topic is limited, no definite mediational hypotheses were formulated in the study.

## Materials and methods

### Design

The cross-sectional study included 382 people with MS who were enrolled with cooperation from two rehabilitation centers in Poland. The study protocol was compliant with the guidelines of the Helsinki Declaration (as revised in Brazil 2013). The inclusion criteria of the study included the following: positive MS diagnosis (McDonald's revised criteria), no relapse during the last 30 days prior to the enrollment, absence of any concomitant disease, and absence of cognitive problems and/or psychiatric disorders as reported by a neurologist. The participants were recruited during the control visit at the rehabilitation clinic by a research team member, between December 2015 and January 2017. After providing verbal instruction and explaining the study purpose, individuals were asked if they were interested to participate. Those who were willing to participate were requested to provide informed consent in written form after assuring them that their data would remain confidential. Participants completed all the questionnaires in a quiet room allotted especially for the needs of this study. If any missing information was found, participants were asked to complete it. Furthermore, a research team member verified the data accuracy. Out of 400 participants, 18 were excluded because they did not complete the questionnaires. Ethical approval for the study was provided by the Bioethical Commission of Poznan University of Medical Science, and the study also complied with the STROBE (Strengthening the Reporting of Observational Studies in Epidemiology) checklist.

### Measures

Self-management was measured with the Polish version of Multiple Sclerosis Self-Management Scale—Revised (MSSM-R) developed by Bishop and Frain [26] and adapted by Tomczak, Kleka and Wilski [27] to perform a multidimensional assessment of the MS patients' behaviors and knowledge about self-management. This scale consists of a total of 24 items. A 5-point scale ranging from 1 (*Disagree completely*) to 5 (*Agree completely*) was used to rate the patient responses. Higher scores signify a greater level of self-management. The reliability as well as validity of MSSM-R was proven in an earlier study [14, 15, 28]. In this study, the MSSM-R scale indicated high internal consistency (Cronbach's $\alpha$ = 0.89).

The perceived level of MS impact was estimated using polish adaptation of Multiple Sclerosis Impact Scale-29 (MSIS-29) [29]. This prominent self-report measure is used for assessing a patient's perception of the impact posed by MS on his/her life in the last 2 weeks [30]. The physical subscale used for assessment included 20 questions evaluating the functional ability of patients, with scores ranging from 20 to 100. Higher scores signify worse perception of patients

about their physical condition. Studies have shown that the MSIS-29 subscale is a sensitive, accurate, and valid tool for assessing MS populations [31, 32]. In this study, the MSIS-29 scale showed high internal consistency (Cronbach's α = 0.98).

Coping strategies were assessed by Endler and Parker's [33] polish version of Coping Inventory for Stressful Situations (CISS) [34]. This is a 48-item measure that evaluates the following styles of coping: problem-oriented, emotion-oriented, and avoidance-oriented. The score for each scale ranges between 16 and 80. The responses are indicated by a Likert-type 5-point scale, ranging from 1 (*Not at all*) to 5 (*Very much*). Higher scores indicate a greater level of coping. The CISS scale is used for assessing different illness groups as its validity and reliability are high [33]. In this study, the CISS scale showed high internal consistency (the α value for problem-oriented, emotion-oriented, and avoidance-oriented style of coping was 0.79, 0.88, and 0.69, respectively).

We used a standardized questionnaire in this study to collect demographic and clinical information of the participants. All participants were requested to provide data about age, sex, education, occupation, marital status, disease duration, and diagnosed MS type. In addition, the neurologist assessed the severity level of MS using Expanded Disability Status Scale (EDSS), which ranges between 0 (*Normal examination*) and 10 (*Death from MS*) [35], and reported whether the participant had any cognitive and/or psychiatric problems.

## Statistical analysis

To determine if the coping strategy used to deal with stress (problem-, emotion-, and avoidance-oriented style of coping) mediates the association between the MS impact (independent variable) and self-management (dependent variable), we performed mediation analysis. Due to the potential importance of disability index (EDSS) and the time since diagnosis of individuals' functioning, these variables were included in the presented model as covariates (especially for self-management level, the correlation with EDSS and time from diagnosis was: r = −0.28, $p<0.001$, bootstrap percentile 95%CI [-.37494; -.18375] and r = −0.27, $p<0.001$, bootstrap percentile 95%CI [-.36420; -.18785] respectively). To determine the significance of indirect effects, we calculated a 95% confidence interval (CI) ~~with~~ using percentile bootstrapping. The statistical analysis was conducted using the PROCESS macro developed by Hayes [36]. In addition to the classical p value generated by macro PROCESS, 95% bootstrap percentile confidence intervals for the path coefficients from the independent variable to the mediators (paths a), and from the mediators to the dependent variable (paths b) were presented in the text. For estimating the bootstrap CI, we used 50000 samples.

## Results

The participants had a mean age of 46 years (126 men and 256 women). With regard to marital status, more than 32% of the participants were single, divorced, or widowed and 67% were married. With regard to employment status, 38,2% were employed and 56% were either retired or were receiving a disability pension. The mean disease duration was 11 years, and the majority of participants had a relapsing–remitting course (41%). The disability level (measured with EDSS) was mild and moderate in more than 90%, and only less than 10% of the participants had a higher level of disability. Table 1 summarizes the sociodemographic and clinical characteristics of the participants.

### Mediation analysis

Statistically significant indirect effects were obtained for problem-oriented coping (-.01384, [95% bootstrap CI: -.03098; -.00054]; standardized effect: -.02831, [95% bootstrap CI: -.06523; -.00107]) and emotion-oriented coping (-.03528, [95% bootstrap CI: -.06084; -.01205];

**Table 1. Sociodemographic and clinical characteristic of the sample.**

| Sociodemographic and clinical characteristic | People with MS(N = 382) | |
|---|---|---|
| Age (years) (mean ± SD; min–max) | 46.4 ± 11.9 (18–82) | |
| Sex n (%) | | |
| Female | 256 | (67.0) |
| Male | 126 | (33.0) |
| Marital status n (%) | | |
| Single | 71 | (18.6) |
| Married | 257 | (67.3) |
| Separated/Divorced/Widowed | 54 | (14.1) |
| Employment n (%) | | |
| Employed | 146 | (38.2) |
| Unemployed | 21 | (5.5) |
| Disability pension/retiring | 215 | (56.3) |
| Time (years) since the diagnosis of MS (mean ± SD; min–max) | 11.6 ± 9.1 (1–48) | |
| Diagnosed type of MS n (%) | | |
| Relapsing–remitting (RRMS) | 158 | (41.4) |
| Primary progressive (PPMS) | 88 | (23.0) |
| Secondary progressive (SPMS) | 70 | (18.3) |
| Progressive–relapsing (PRMS) | 33 | (8.6) |
| Don't know (DKMS) | 33 | (8.6) |
| Disability subgroups "EDSS" n (%) | | |
| Mild (EDSS ≤ 3.5) | 166 | (43.5) |
| Moderate (3.5 < EDSS ≤ 6.5) | 181 | (47.4) |
| High (EDSS < 6.5) | 35 | (9.2) |

Abbreviations: EDSS, expanded disability status scale; SD, standard deviation; MS, multiple sclerosis; CI, confidence interval.

standardized effect: -.07219, [95% bootstrap CI: -.12486; -.02470]) (Table 2). A negative association was found between the MS impact and problem-oriented coping (B = −0.07, $p<0.001$; [95% bootstrap CI:-.09689; -.03717 ]), while a positive association was found between problem-oriented coping and individual's self-management (B = 0.20, marginally significant effect based on the classical p value: $p = 0.0539$; however, 95% bootstrap CI was: [.00884; .39312]). In addition, a positive association was found between the MS impact and emotion-oriented coping (B = 0.16, $p<0.001$; [95% bootstrap CI: .12694; .19695 ]), and a negative association was found between emotion-oriented coping and individual's self-management (B = −0.22, $p<0.01$; [95% bootstrap CI: -.36168; -.07521]) (Fig 1).

**Table 2. Indirect effects for mediational relations between the impact of the disease, coping strategies, and self-management in people with MS.**

| Potential mediator | Indirect effect [95% CI] | Standardized indirect effect [95% CI] |
|---|---|---|
| Problem-oriented coping | -.01384 [-.03098; -.00054] | -.02831 [-.06523; -.00107] |
| Emotion-oriented coping | -.03528 [-.06084; -.01205] | -.07219 [-.12486; -.02470] |
| Avoidance-oriented coping | -.01097 [-.02776; .00012] | -.02245 [-.05550; .00024] |
| **Total** | -.06009 [-.09138; -.03150] | -.12295 [-.18813; -.06417] |

Abbreviations: MS, multiple sclerosis; CI, confidence interval.

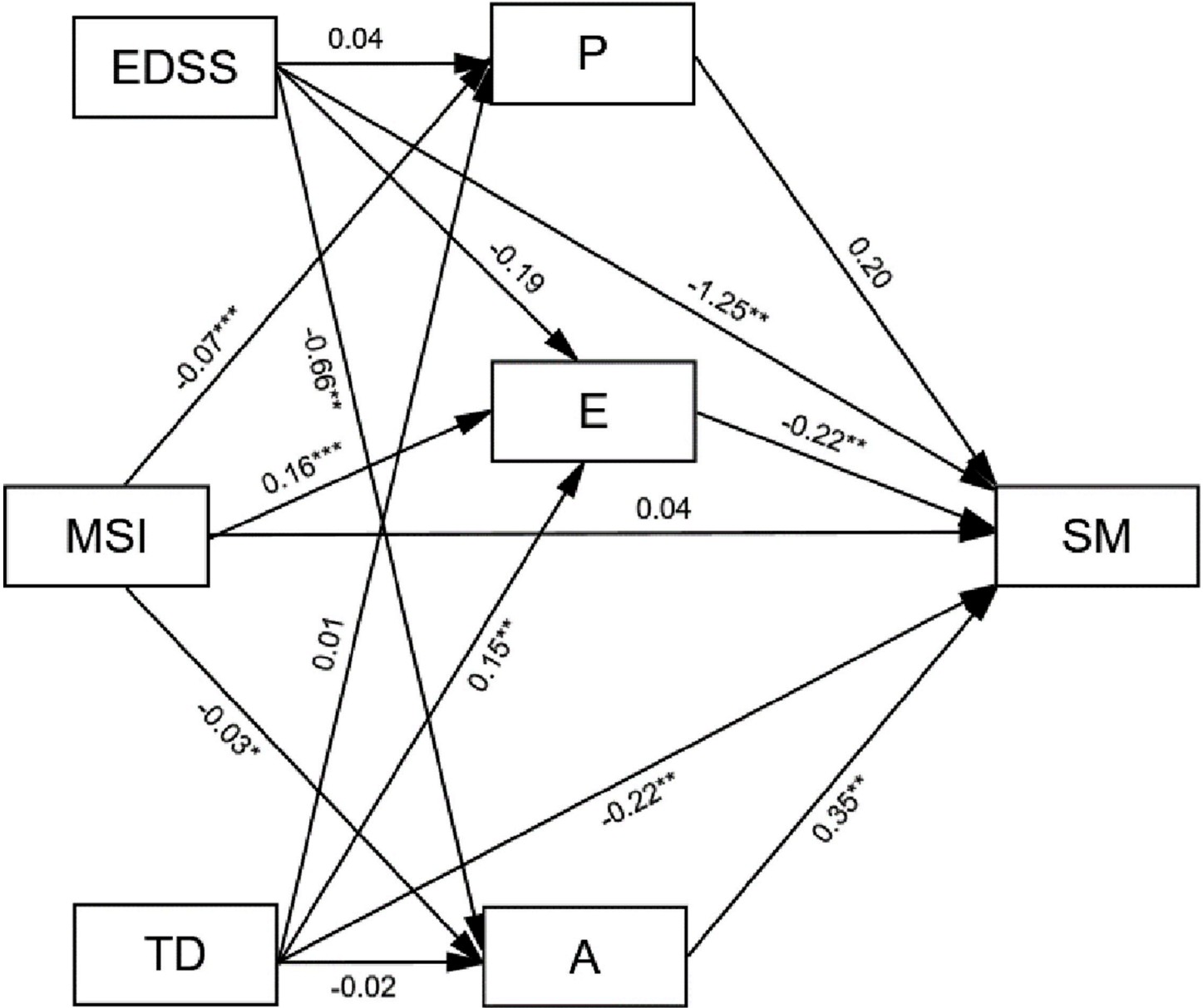

**Fig 1.** Coping strategies [problem-oriented (P), emotion-oriented (E), and avoidance-oriented (A)] as mediators between the impact of multiple sclerosis (MSI) and individuals' self-management (SM) with expanded disability status scale (EDSS) and time from diagnosis (TD) as covariates.

Indirect effect for avoidance-oriented coping did not achieve statistical significance: -.01097, [95% bootstrap CI: -.02776; .00012]; standardized effect: -.02245 [95% bootstrap CI: -.05550; .00024] (Table 2). However, the path coefficients between MS impact and avoidance-oriented coping and between avoidance-oriented coping and individual's self-management were significant both in the light of the classical p value as well as the bootstrap method (MS impact—avoidance-oriented coping: B = -. 03, p < .05; [95% bootstrap CI: -.05924; -.00057]; avoidance-oriented coping—individual's self-management: B = .35, p <0.01; [95% bootstrap CI: .09121; .62305]). Both total (B = −0.02, $p > 0.05$) and direct effects (B = 0.04, $p > 0.05$) between the MS impact and individual's self-management did not reach statistical significance.

## Discussion

As far as we know, this study was the first to analyze whether coping strategies act as mediators in the association between the perceived MS impact and the level of self-management in people with MS. Our results provide the following conclusions. First, no significant relationship was found between the perceived MS impact and the level of self-management under control of time from diagnosis and disability level. Although there was no primary relationship between the independent and dependent variable, new standards in the mediation analysis indicate that this relationship is not necessary (not required). The currently conducted mediation analysis emphasizes only a statistically significant indirect effect and its substantive justification [36–39]. Second, individuals with a worse perception of MS used more emotion-coping strategies, and this was associated with lower level of self-management under control of time from diagnosis and disability level. Third, a negative association was found between the MS impact and problem-oriented coping, while a positive association was observed between problem-oriented coping and self-management under control of time from diagnosis and disability level. Finally, the mediating effect was not observed for avoidance-oriented coping.

According to our findings, in the case of MS, perception of the disease does not directly translate into an individual's activity in the form of self-management; however, a part of this relationship is mediated by coping practices. This means that the individual's perception of the MS severity guides him/her to take up a coping strategy, which in turn is related to self-management activity. This agrees with the speculative assumptions of the Common-Sense Model of Self-Regulation.

Specifically, our results indicate that people with MS with a worse perception about the MS impact tend to apply emotion-oriented coping strategies, which in turn is associated with low level of self-management. This seems reasonable as earlier studies have demonstrated that health deterioration of an MS patient is associated with depression and due to applying less adaptive emotion-centered coping [40]. Other studies have reported that higher disability in people with MS is linked with high scores on depression scales and these people display less flexible, emotion-focused strategies of coping [41]. This clearly shows that worse perception about the impact of MS negatively influences the psychological health of people with MS. As a result, the patients apply emotional strategies, which are negatively associated with their adjustment [42], quality of life [43], and level of self-management (as revealed in this study).

Problem-oriented coping also mediated the association between the perceived MS impact and the level of self-management. These coping strategies are generally considered to be more adaptive and by definition are related to active participation in the treatment process. Our study showed that people with MS who have a more favorable perception of the disease use more adaptive problem-coping strategies, which was related to more effective self-management. However, empirical evidence from earlier studies has revealed mixed findings. For example, in a previous study, problem-oriented coping strategies were unrelated to adjustment indicators [44], whereas in another study, the results showed that these strategies were related to better psychological health [21, 25]. Some researchers indicate that in the case of conditions that could not be controlled by behavioral changes or treatment, problem-oriented coping is of no value [45–47]. However, our study corroborates the hypothesis that a greater focus on problem-oriented coping strategies is associated with higher level of self-management in people with MS.

Despite the significant relationships between the avoidance-oriented coping, perceived impact of MS, and level of self-management, no significant indirect effect was observed, which indicates that avoidance is not a mediator in the analyzed relationship. The efficacy of avoidance-oriented coping in scientific research remains unclear due to divergent results. Most of

the findings report that avoidance-oriented coping is maladaptive, which translates to worse adjustment in people with MS [48, 49]; however, some studies indicate that avoidance can be applied as a kind of adaptive approach for a certain period to avert negative perception about future MS progression and prevent depression [23, 50, 51]. A possible explanation of this contradiction might be that there are differences in the definition of avoidance-oriented coping. For example, in the case of the frequently used Brief COPE questionnaire, avoidance means denial, behavioral disengagement, and substance use. In our study, we used the CISS questionnaire, which defines avoidance as activities or cognitive modifications adapted to avoid a stressful circumstance, which is achieved through diverting to other situations or engaging oneself in or other tasks (task-oriented) or by social diversion (person-oriented) for reducing stress. Such defined avoidance does not have to be nonadaptive. Invariably, our results do not clarify the role of avoidance in shaping self-management, which certainly needs further in-depth research.

In addition, our results provide evidence that the level of self-management decreases with higher disability and a longer duration of the illness. This is contradictory to our earlier reports [14, 15], but the difference may stem from the variations in the examined groups. Our study sample included a significantly higher percentage of participants with mild EDSS (≤3.5; 43%) than that of previous studies (11% and 8%, respectively). There is a need for research on a more representative group in terms of disability, which may explain the discrepancy in results.

It is important to emphasize that the theoretical background of these studies is based on the Common-Sense Model of Self-Regulation, which assumes that disease perception is related to self-management activities through coping strategies that serve as mediators [18]. Such an assumption implies certain conclusions, however, to fully confirm our interpretation, longitudinal models of testing are needed. Despite the need for longitudinal studies for testing causality, the results of our study are of practical value to people with MS as well as health professionals dealing with them. First, people with MS and health professionals should be educated on the possible effect of negative illness perception and emotion coping strategies on self-management. Second, actions directed at ways of coping should be indicated, because they mediate the level of self-management. Special attention should be paid to lowering the emotional techniques of coping and motivating individuals to engage in problem-solving coping. Although these relationships are known in relation to other forms of activity of people with various diseases, the present study for the first time indicates their importance in self-management of people with MS, which is an uncertain, unpredictable, and difficult-to-manage disease. Third, appraisal of MS impact needs to be regularly examined in individuals, particularly in those showing poor involvement in the treatment process. According to our assumptions, exaggerating the risk and complications of MS might lead to choosing less adaptive, more emotion-oriented coping strategies and, as a consequence, reduce self-management activities. Challenging the negative perception of MS impact and developing a positive perception could promote more adaptive forms of coping. This can be achieved through offering objective data on the characteristics of the illness and its progression, as well as promoting cognitive changes by eliminating negative beliefs and strengthening positive thoughts. Due to frequent contact with patients, medical personnel play a special role in this respect. These interventions could help in the proposal of patient-focused self-management approaches and be advantageous to medical programs treating people with MS.

The present study has some limitations which should be taken into account when interpreting the results. First, this is a cross-sectional study, which does not allow any conclusion to be made regarding the directionality of the associations. Therefore, longitudinal studies should be carried out to provide additional insights on the mediation patterns

reported here. Second, the data examined in the study were self-reported, which indicates the possibility of response bias as well as shared method bias. Third, the majority of participants (91%) showed mild and moderate disability (EDSS ≤6.5), while only 9% showed severe disability, which limits the possibilities about the generalization of the results. In addition, we excluded individuals with cognitive problems through subjective neurological examination unsupported by an objective research instrument. This can undermine the applied exclusion criterion. Finally, we cannot exclude other variables that have a potential relationship with the level of self-management in people with MS. Further research should be focused on identifying those factors whose modifications will allow developing self-management skills.

## Conclusion

The findings of our study highlight coping strategies as a modifiable factor of the association between the perceptions of MS impact and the level of self-management. Specifically, with a worse perception of the disease, people with MS tend to use more emotion-centered coping, which is negatively associated with their self-management. On the other hand, with a better perception of the disease, individuals use more problem-oriented strategies, which is associated with a higher level of self-management. Thus, lowering the emotional response and increasing problem-solving attitude promote activities related to the involvement of people with MS in the process of treatment and rehabilitation.

## Supporting information

**S1 Data.**
(XLSX)

## Acknowledgments

We would like to gratefully acknowledge all the individuals with multiple sclerosis who completed a questionnaire and Dr. Mariusz Kowalewski, the manager of Multiple Sclerosis Rehabilitation Centre in Borne Sulinowo for his help in enrollment of the study participants.

## Author Contributions

**Conceptualization:** Maciej Wilski.

**Data curation:** Maciej Wilski, Waldemar Brola, Maciej Tomczak.

**Formal analysis:** Maciej Wilski, Magdalena Łuniewska, Maciej Tomczak.

**Methodology:** Maciej Wilski, Magdalena Łuniewska, Maciej Tomczak.

**Project administration:** Maciej Wilski, Waldemar Brola.

**Resources:** Waldemar Brola.

**Supervision:** Waldemar Brola, Magdalena Łuniewska.

**Validation:** Maciej Tomczak.

**Writing – original draft:** Maciej Wilski, Maciej Tomczak.

**Writing – review & editing:** Maciej Wilski, Magdalena Łuniewska.

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
