## [Decision Letter · Decision Letter 0]

10 Dec 2020

PONE-D-20-29426

Impact of multiple sclerosis and self-management: mediating role played by coping strategies

PLOS ONE

Dear Dr. Wilski,

Thank you for submitting your manuscript to PLOS ONE. After careful consideration, we feel that it has merit but does not fully meet PLOS ONE’s publication criteria as it currently stands. Therefore, we invite you to submit a revised version of the manuscript that addresses the points raised during the review process.

Many thanks to the reviewers for their valuable comments on the manuscript. Reviewer 1 raised important concerns about the analytic approach that must be addressed, certainly with clearer presentation of your results and possibly with an entirely new analysis approach, as they suggest. In addition, I am concerned about the cross-sectional nature of your analyses. You note this as a weakness in the discussion, but then go on to interpret the implications in causal terms. The data cannot support these conclusions. For example, your correlations do not rule out the possibility that coping is an outcome of self-management rather than vice versa. Instead a longitudinal cross-lagged mediation model would be more appropriate to test the relationships that are discussed in the manuscript. Please revise your discussion to make this limitation clearer and to explicitly consider the alternative causal chains that might underlie your results and the potential implications each of these might have.

We look forward to receiving your revised manuscript.

Kind regards,

Richard Rowe

Academic Editor

PLOS ONE

Journal Requirements:

Reviewers' comments:

Reviewer's Responses to Questions

**Comments to the Author**

1. Is the manuscript technically sound, and do the data support the conclusions?

Reviewer #1: No

Reviewer #2: Yes

2. Has the statistical analysis been performed appropriately and rigorously? 

Reviewer #1: No

Reviewer #2: Yes

3. Have the authors made all data underlying the findings in their manuscript fully available?

Reviewer #1: Yes

Reviewer #2: Yes

4. Is the manuscript presented in an intelligible fashion and written in standard English?

Reviewer #1: No

Reviewer #2: No

5. Review Comments to the Author

Reviewer #1: First I would like to thank you for allowing me to review this paper, which has the potential to contribute to the MS literature. The topic is novel and interesting. However, it seems to me that the study has some shortcomings in terms of data analytical strategy. I have included specific comments made on the manuscript, which I am submitting as an attachment.

Reviewer #2: The study is interesting and I think that the results could add some important points in the clinical and therapeutic managment of patient with MS increasing their involvement in the care process.

I do not suggets modification of specific points and I do not see criticisms related to the methods and to the interpretations of the results.

Minor point:

1) line 49 Multiple Sclerosis (MS) - (use capital letter also for Sclerosis).

2) I suggest using "Person with Multiple Sclerosis (PwMS)" rather than patient throughout the manuscript.

6. PLOS authors have the option to publish the peer review history of their article (what does this mean?). If published, this will include your full peer review and any attached files.

Reviewer #1: No

Reviewer #2: No

---

## [Author Response · Author response to Decision Letter 0]

26 Jan 2021

Response to Reviewers

“The perceived impact of MS and self-management: The mediating role of coping strategies”

Based on the valuable guidelines we have made some changes in the text. We hope that the end result is satisfactory for the reviewers and editors. 

Editor:

We are very grateful for all the suggestions. We have improved the text in accordance with most of the instructions.

1. I am concerned about the cross-sectional nature of your analyses. You note this as a weakness in the discussion, but then go on to interpret the implications in causal terms. The data cannot support these conclusions. For example, your correlations do not rule out the possibility that coping is an outcome of self-management rather than vice versa. Instead a longitudinal cross-lagged mediation model would be more appropriate to test the relationships that are discussed in the manuscript. Please revise your discussion to make this limitation clearer and to explicitly consider the alternative causal chains that might underlie your results and the potential implications each of these might have.

Thank you for your suggestion. In fact, despite the caution, the text contains statements that suggest a cause-and-effect relationship. After careful analysis, we removed these statements. We hope you are satisfied with the end result. We have also made some changes to the analysis of mediation, which in our opinion correspond to the latest recommendations and are currently the standard in this type of analysis (we describe it precisely in responses to the comments of the reviewer).

According to longitudinal cross-lagged mediation model we cannot apply this analysis because we have not done longitudinal studies. The time from the diagnosis day was measured at one point in time as an additional variable simply to assess the duration of the disease.

Reviewer #1:

1. I suggest to revise the title. A possible alternative could be "The perceived impact of MS and self-management: The mediating role of coping strategies"

Thank you for your suggestion. Changes have been made.

2. Please decide which term you will use and stick to it throughout the entire paper. In accordance with ethical guidelines, I suggest "people with MS". Prior sentences include "patients with MS" and "MS patients", resulting in an inconsistency.

 According to your suggestion we have decided to use “people with MS”.

3. Are you sure its the "main element"? I suggest using a less ambitious adjective.

Thank you for your suggestion. Changes have been made.

4. Was this scale tested for validity and reliability in the Polish MS population before? If yes, please cite the study which carried out the Polish adaptation of the scale.

Polish validations of all scales have been included in the text.

5. You used "sex" before, so please decide which term to use. I suggest "sex" since you seem to have questioned the biological sex of the participants.

Thank you for your suggestion. Changes have been made.

6. According to Baron & Kenny, the independent variable has to significantly predict the dependent variable prior to testing any mediation. In your study, there was no significant association between MS impact and self-management. Maybe this lack of association stems from the use of control variables. Since there was no significant association between your dependent and independent variables, I don't think that it is possible to carry on with the mediation analysis. I believe that your mediation model needs to be revised and maybe you can employ a different data analytic approach to test the proposed model. I can even suggest changing your data analysis completely and carry out a multiple regression instead.

Indeed, there was no primary relationship between the independent and dependent variables as suggested by Baron and Kenny. However, nowadays in the mediation analysis this relationship is not necessary (not required). The currently conducted mediation analysis emphasizes only a statistically significant indirect effect and its substantive justification (e.g. Hayes 2017, Hayes 2013, Rucker et al. 2011, Zhao et al. 2010). In order to verify the significance of the mediation effect, the percentile bootstrap is suggested.

Hayes, A. F. (2013). Introduction to mediation, moderation, and conditional process analysis: A regression-based approach. Guilford publications.

Hayes, A. F. (2017). Introduction to mediation, moderation, and conditional process analysis: A regression-based approach. Guilford publications.

Zhao, X., Lynch Jr, J. G., & Chen, Q. (2010). Reconsidering Baron and Kenny: Myths and truths about mediation analysis. Journal of consumer research, 37(2), 197-206.

Rucker, D. D., Preacher, K. J., Tormala, Z. L., & Petty, R. E. (2011). Mediation analysis in social psychology: Current practices and new recommendations. Social and Personality Psychology Compass, 5(6), 359-371.

Below we quote Hayes' reply from his website (source: http://processmacro.org/faq.html): 

„Question: It appears that I have evidence of an indirect effect of X on Y through a proposed mediator, but there is no evidence of an association between X and Y. Is this possible? What should I do?

Answer: This is not only possible, but it is probably much more common than people realize. Modern thinking about mediation analysis does not impose the requirement that there be evidence of a simple association between X and Y in order to estimate and test hypotheses about indirect effects. See Hayes, A. F.(2009). Beyond Baron and Kenny: Statistical mediation analysis in the new millennium. Communication Monographs, 76, 408-420. [PDF] or Hayes, A. F., & Rockwood, N. J. (2017). Regression-based mediation and moderation analysis in clinical research: Observations, recommendations, and implementation. Behaviour Research and Therapy, 98, 39-57. [PDF]. Also see Chapter 4 of Hayes (2018) or Chapter 15 of Darlington and Hayes (2017).”

Additionally, as for the correlation between covariates and the dependent variable, and for mediation indirect effects, we added bootstrap confidence intervals for mediation path coefficients "a" and "b" in the text. These are often recommended because they inform us in which interval is the real parameter. In addition, the bootstrap CI is more robust than the standard p value. In our case, when it comes to the level of 0.05, there is one difference: the relation between the problem copying and the dependent variable was marginally significant, while based on the bootstrap method it is slightly significant. However, this does not matter for our results, according to Hayes, because we draw conclusions for the mediation effects based on the confidence intervals for the indirect effects only. In order to obtain intervals for the coefficients we repeated the analysis again, which gave slightly different confidence intervals for indirect effects (this is natural for the bootstrap method).

7. This title is too long.

In our opinion, the title shows the full content of the figure without having to add a note. Of course we can change it, we leave the decision to the editor.

Reviewer #2:

Minor point:

1) line 49 Multiple Sclerosis (MS) - (use capital letter also for Sclerosis).

2) I suggest using "Person with Multiple Sclerosis (PwMS)" rather than patient throughout the manuscript.

Changes have been made according to your suggestions.

---

## [Editor Report · Decision Letter 1]

1 Feb 2021

PONE-D-20-29426R1

The perceived impact of multiple sclerosis and self-management: The mediating role of coping strategies

PLOS ONE

Dear Dr. Wilski,

Thank you for submitting your manuscript to PLOS ONE. After careful consideration, we feel that it has merit but does not fully meet PLOS ONE’s publication criteria as it currently stands. Therefore, we invite you to submit a revised version of the manuscript that addresses the points raised during the review process.

Many thanks for your revisions to this ms. I believe it requires a few more minor modifications before it can be published:

P5 line 100 shown not proven

P6 line 129 "the Polish”

P11 Line 236. You provide a good answer to Reviewer 1’s question about conducting mediation analysis. I think this issue may be in the minds of other readers too. So please very briefly note that mediation is an appropriate analysis approach

P13 Line 296 testing, not proving

While the language on causality may have been revised, the interpretation in this paragraph still assumes causality based on theoretical grounds. For example, it is suggested that "people with MS and health professionals should be educated on the effect of negative illness perception and emotion coping strategies on self-management.” This assumes a direction of effect, when it could be, for example, that coping strategy drives illness perception rather than vice versa. It is fine to base the interpretation on theoretical grounds, but it needs to be clear that is what you are doing and that longitudinal models will be helpful in testing whether that interpretation is indeed correct. So please consider revising the discussion to address this point further.

We look forward to receiving your revised manuscript.

Kind regards,

Richard Rowe

Academic Editor

PLOS ONE

---

## [Author Response · Author response to Decision Letter 1]

18 Feb 2021

Response to Reviewers

“The perceived impact of MS and self-management: The mediating role of coping strategies”

Based on the valuable guidelines we have made some changes in the text. We hope that the end result is satisfactory for the reviewers and editors. 

Editor:

P5 line 100 shown not proven

P6 line 129 "the Polish”

P11 Line 236. You provide a good answer to Reviewer 1’s question about conducting mediation analysis. I think this issue may be in the minds of other readers too. So please very briefly note that mediation is an appropriate analysis approach

P13 Line 296 testing, not proving

While the language on causality may have been revised, the interpretation in this paragraph still assumes causality based on theoretical grounds. For example, it is suggested that "people with MS and health professionals should be educated on the effect of negative illness perception and emotion coping strategies on self-management.” This assumes a direction of effect, when it could be, for example, that coping strategy drives illness perception rather than vice versa. It is fine to base the interpretation on theoretical grounds, but it needs to be clear that is what you are doing and that longitudinal models will be helpful in testing whether that interpretation is indeed correct. So please consider revising the discussion to address this point further

We are very grateful for all the suggestions. We have improved the text in accordance with all of your instructions.

---

## [Editor Report · Decision Letter 2]

22 Feb 2021

The perceived impact of multiple sclerosis and self-management: The mediating role of coping strategies

PONE-D-20-29426R2

Dear Dr. Wilski,

Very many thanks for making the latest round of revisions so effectively. We’re pleased to inform you that your manuscript has been judged scientifically suitable for publication and will be formally accepted for publication once it meets all outstanding technical requirements.

Kind regards,

Richard Rowe

Academic Editor

PLOS ONE
---

## [Editor Report · Acceptance letter]

24 Feb 2021

PONE-D-20-29426R2 

The perceived impact of multiple sclerosis and self-management: The mediating role of coping strategies 

Dear Dr. Wilski:

I'm pleased to inform you that your manuscript has been deemed suitable for publication in PLOS ONE. Congratulations! Your manuscript is now with our production department. 

Kind regards, 

on behalf of

Professor Richard Rowe 

Academic Editor

PLOS ONE